# Effect of Ozone-Treated or Untreated Saskatoon Fruits (*Amelanchier alnifolia* Nutt.) Applied as an Additive on the Quality and Antioxidant Activity of Fruit Beers

**DOI:** 10.3390/molecules27061976

**Published:** 2022-03-18

**Authors:** Józef Gorzelany, Dorota Michałowska, Stanisław Pluta, Ireneusz Kapusta, Justyna Belcar

**Affiliations:** 1Department of Food and Agriculture Production Engineering, Collegium of Natural Sciences, University of Rzeszow, 4 Zelwerowicza Street, 35-601 Rzeszów, Poland; gorzelan@ur.edu.pl; 2Laboratory of Beer and Malt, Department of Food and Vegetable Product Technology, Prof. Wacław Dąbrowski Institute of Agricultural and Food Biotechnology—State Research Institute, 36 Rakowiecka Street, 02-532 Warsaw, Poland; dorota.michalowska@ibprs.pl; 3Department of Horticultural Crop Breeding, The National Institute of Horticultural Research, Konstytucji 3 Maja 1/3 Street, 96-100 Skierniewice, Poland; stanislaw.pluta@inhort.pl; 4Department of Food Technology and Human Nutrition, Collegium of Natural Sciences, University of Rzeszow, 4 Zelwerowicza Street, 35-601 Rzeszów, Poland; ikapusta@ur.edu.pl

**Keywords:** Saskatoon fruits, ozonation, fruit beers, beer quality, antioxidant potential of beer

## Abstract

Fruit of Saskatoon (*Amelanchier alnifolia* Nutt.) are a good source of bioactive compounds, such as polyphenols, including anthocyanins, as well as vitamins, macro- and microelements and fibre. By treating Saskatoon fruits with gaseous ozone, and adding the material as an enhancer to barley beers, it is possible to impact the contents of bioactive compounds in the produced fruit beers. Sensory tests showed that beers made from barley with addition of Saskatoon fruit of the ‘Smoky’ cultivar were characterised by the most balanced taste and aroma. Physicochemical analyses of fruit beers, produced with Saskatoon fruit pulp added on the seventh day of fermentation, showed that the beers enhanced with ozone-treated and untreated ‘Smoky’ Saskatoon fruits had the highest contents of alcohol, 5.51% *v*/*v* and 5.66% *v*/*v*, respectively, as well as total polyphenol contents of 395 mg GAE/L and 401 mg GAE/L, respectively, and higher antioxidant activity (assessed using DPPH^•^, FRAP and ABTS^+•^ assays). It was demonstrated that the ozonation process led to a decrease in the contents of neochlorogenic acid, on average by 91.00%, and of caffeic acid by 20.62%, relative to the beers enhanced with ‘Smoky’ Saskatoon fruits not subjected to ozone treatment. The present study shows that Saskatoon fruits can be used in the production of beer, and the Canadian cultivar ‘Smoky’ is recommended for this purpose.

## 1. Introduction

Traditionally, beer is an alcoholic beverage produced from water, malt, hop and yeast. In recent years, however, consumers have shown increasing interest in beers with added fruit, juice or concentrates and fruit aromas. Breweries started production of Radler beer, i.e., a combination of beer and, most commonly, fruit juice or flavoured sugar syrup; this low-alcohol drink with a distinctive flavour is particularly attractive to female consumers [1]. Other examples of fruit beer include the typical Belgian Lambic beer, a result of spontaneous fermentation, produced with the addition of cherries ‘Kriek’ or raspberries ‘Framboise’ [2,3,4]. The fruit is most commonly added during fermentation of young beer or during refermentation in bottles. The most commonly used raw materials include raspberries, cherries, strawberries, bananas, forest berries or exotic fruit. Fruit beers are characterised by the colour, taste and aroma of the specific type of fruit; they also have a greater antioxidant potential and higher contents of bioactive compounds passing to the beer from the fruit, as well as fine refreshing properties [2,5]. Fruit beers produced in craft breweries using very simple methods provide an impulse for changes in the beer market. These types of beer present very good sensory properties as a result of small-scale brewing technologies and production based on unique recipes, which leads to a higher quality of the final product [2,4].

The Saskatoon shrub (*Amelanchier alnifolia* Nutt.) produces purple fruit, oval in shape, and resembling “small apples”. In Poland, it is still a fairly unknown species, grown only in small-size farms. Saskatoon for years has been mainly cultivated in Canada, because of the valuable fruit [6,7,8]. Saskatoon fruits are a good source of phenols, including anthocyanins, flavonoids, minerals, vitamins, lipids and dietary fibre [9,10,11]. These compounds, of high nutritional value, are beneficial for people. Furthermore, the bioactive compounds contained in Saskatoon fruits show antioxidant, antiatherosclerotic and bactericidal properties. Moreover, raw or processed Saskatoon fruits beneficially affect the cardiovascular system, reduce blood pressure and improve vision [12,13]. These fruits may be consumed as fresh and processed or frozen (IQF).

The contents of nutritional compounds in Saskatoon fruits depend on the maturity, as well as growing and harvesting conditions, the genotype and storage conditions [14,15]. In the food processing industry, Saskatoon fruits are used in the production of beverages, syrups, wine, jam or jelly; they can also be added as an ingredient to pies or cakes [11,15,16]. They may be mixed with other fruit, particularly those with a sour taste. Saskatoon syrup may be used in the production of functional beverages because of its stability, as well as its antioxidant properties, flavour and colour [17]. Saskatoon fruits are used as natural food colorants due to the high content of anthocyanins [18]. Thanks to the high antioxidant potential of the phenolic compounds contained in Saskatoon fruits, it is possible to use Saskatoon pomace (a waste in production of juice) as a natural source of antioxidants, to extend the shelf-life of food products, while improving their quality, taste, colour, aroma, texture and appearance [19].

The storage shelf-life of fruit may be improved by applying ozonation. Ozone is a chemical medium with strong oxidizing properties, which also decomposes very quickly into oxygen. Studies have shown that the use of ozone in gaseous form is more effective compared with the aqueous form of ozone (faster process of ozone breakdown in water). By applying ozone, it is possible to limit the development of various fungal diseases, such as grey mould (*Botrytis cinerea*), which has an impact on extending the shelf life of fruits, especially those used in food processing (ozone can be used at any stage—immediately after harvest, during transport, sorting or packaging). Ozonation leads to improved processing properties (including reducing ethylene secretion, increasing polyphenol content and antioxidant activity in fruits, and causing modifications in the activity of enzymes found in vegetables) and microbiological safety of food [20,21,22,23,24]. Fruits have microorganisms on their surface, especially yeasts and moulds, which during fermentation pass into young beer, causing undesirable processes that significantly affect both the appearance of beer (colour) as well as taste sensations (e.g., the formation of diacetyl or phenols). The applied process of ozonation of fruits before their addition to beer can inhibit the growth of microorganisms affecting the microbiological stability of the finished product.

The purpose of this study was to identify the physicochemical, sensory and antioxidant properties of beers produced with addition of ozone-treated or untreated Saskatoon fruits. The study also aimed to analyse the findings to determine the feasibility of three Saskatoon fruit cultivars for the production of fruit beers.

## 2. Results and Discussion

### 2.1. Physicochemical Characteristics of the Beers

The findings related to the physicochemical parameters of barley beers produced with addition of pulp from Saskatoon fruits of three cultivars, treated with ozone or untreated, are shown in Table 1.

The contents of apparent extract in the beer samples ranged from 2.59 to 3.29% *m*/*m*. The highest contents were identified in two beer samples, produced with theaddition of ozone-treated ‘Amela’ fruits (AB0) and ‘Martin’ fruits (MB0); these were significantly higher than the values observed in the control (CB) sample (2.74% *m*/*m*). The contents of real extract in the beer samples were higher than those of apparent extract, and the values ranged from 4.57 to 4.96% *m*/*m*. The contents of original extract varied across the beer samples (11.87–13.32% *m*/*m*). The largest value was identified in the beers produced with addition of ‘Smoky’ fruit (SB0 and SB1; Table 1).

The degree of final attenuation corresponds to the content of ethyl alcohol in beer, i.e., one of the major determinants of quality, as it is responsible for the sensory characteristics of beer [25]. Fruit beers should have higher contents of ethanol compared to beer which is not enhanced with fruit [26]. Out of all the beer samples assessed, the highest alcohol contents and the highest degree of final real attenuation were identified in the beer produced with addition of ‘Smoky’ fruits, whether or not treated with ozone. The remaining beers enhanced with Saskatoon pulp, were found with ethyl alcohol contents below 5.3%, which was less than in CB samples. Notably, the beers produced with addition of ozone-treated fruit had higher contents of ethanol compared to the beers enhanced with untreated fruit pulp. A significantly higher degree of final real attenuation was found in the beer samples enhanced with ozone-treated fruit pulp, irrespective of the Saskatoon cultivar used (Table 1). The energy value identified in the beers produced with addition of the Saskatoon fruits was slightly lower compared to CB, with an exception of the beers enhanced with ‘Smoky’ fruits (Table 1). Beers produced with addition of mango juice and pulp were found with calorific value of 34.13–36.73 kcal, alcohol contents of 4.13–4.27%, and the degree of final true fermentation in the range of 71.30–74.22% [26]. Yang et al. [5] found 3.5% content of alcohol in cherry and blueberry beer samples compared to 2.5% content in raspberry beer. A study by Patraşcu et al. [1] reported 2.8–3.5% ethanol content in raspberry beer, and other researchers found fruit beer samples with alcohol content in the range of 5–8% [3]. Furthermore, Baigts-Allende et al. [2] reported that beer with added cherries had alcohol content in the range of 3.2–8.0%, compared to samples with the added raspberries, 2.5–5.7%, and blackcurrants, 7.1%.

A significantly higher colour index was identified in beer enhanced with fruit pulp compared to the CB sample. The largest values were found in the beers produced with addition of ‘Smoky’ fruits, whether or not they were treated with ozone, and in the sample of beer with addition of ozone-treated ‘Amela’ pulp (AB1) (Table 1, Figure 1, Figure 2 and Figure 3). Both the dose used and the exposure time of ozone affect the colour of the fruits undergoing the process. Ozonated fruits retain their natural colour longer that non-ozonated fruits, which darken faster [20,21,22,23,24]. Already at the stage of preparation of the Saskatoon fruit pulp, a difference in colour between ozonated and non-ozonated fruits within the cultivar was noticed, which had an impact on the final colour of the resulting fruit beers. In the study by Baigts-Allende et al. [2], beer with added blackcurrant was found to have a colour index of 14.97, whereas the value reported by Patraşcu et al. [1] for raspberry beer amounted to 21.16.

The acidity of beers enhanced with Saskatoon fruit was slightly higher than in control (CB) sample. On the other hand, the pH values in the fruit beers were in the narrow range of 4.42–4.61 (Table 1). Likewise, the contents of carbon dioxide were similar in the samples, ranging from 0.43% to 0.47%. In the study by Patraşcu et al. [1], raspberry beers had CO_2_ contents in the range of 0.55–0.65%, as well as acidity and pH values amounting to 2.84–3.50 and 4.24, respectively. Fruit beers investigated by Nardini and Garaguso [3] were found with pH in the range of 3.56–4.86. Lower pH positively affects the microbiological stability of beer by reducing growth of unwanted flora [26]. The addition of fruit on the seventh day of fermentation results in greater microbiological risk, consequently, application of ozone treatment to the fruit material seems justified.

The highest contents of bitter substances were identified in the control (CB) sample. Fruit beers generally were less bitter, however, addition of ozone-treated Saskatoon fruit led to a greater decrease in the contents of bitter substances in the beer, compared to the samples produced with addition of untreated fruit (Table 1). The contents of bitter substances in beer are significantly related to the dose and variety of hops applied, and the degree of isomerisation of hop α-acids, which results from the duration of boiling and, to a lesser extent, from the reaction of proteins with polyphenols contained in the malt [27,28]. The addition of Saskatoon fruits, characterised by high content of sugars (on average, 14.78 g 100 g^−1^ d.m., depending on Saskatoon cultivar [9]), does not only affect the fermentation process, but also results in a decreased bitter taste, which is characteristic for beer products.

### 2.2. Volatile Organic Compounds of the Beers

The contents of volatile organic compounds in beer significantly affect its flavour. Higher than recommended contents of the compounds, e.g., esters or higher alcohols referred to as fuselols, adversely affect the desirability of beer products for customers. Fruit beers produced in this study had low contents of acetic aldehyde, isoamyl acetate and ethyl acetate, except for the beer enhanced with untreated ‘Martin’ fruits (Table 2; sensory thresholds for volatile organic compounds: acetic aldehyde 20–25 mg/L, ethyl acetate 30 mg/L and isoamyl acetate 0.5–1.7 mg/L [29,30]. Ethyl acetate exceeding normative levels produces a characteristic peach-like, flat and excessively sweet aftertaste, and in many types of beer, its contents of up to 80 mg/L have been reported [29,31,32]. The contents of n-propanol and amyl alcohols identified in fruit beers also correspond to normative values (Table 2; sensory thresholds for: n-propanol 4–48 mg/L, isobutanol 4–57 mg/L and isoamyl alcohol 25–123 mg/L [29,30]). On the other hand, the contents of isobutanol in all the assessed beers produced with addition of Saskatoon fruit pulp and in control (CB) samples exceeded the normative values (Table 2). Fusel alcohols, including n-propanol and isobutanol, are responsible for alcoholic and solvent-like aroma, and they are frequently perceived as pungent in taste. The lowest content of diacetyl was found in the CB sample; however, all the fruit beer samples, except for MB1, were found with acceptable contents of the compound (<25 µg/L; Table 2). Diacetyl (butane-2,3-dione) is a side product affecting the flavour of beer [33].

### 2.3. Contents of Bioactive Compounds in Fruit Beers

Saskatoon fruits are a good source of polyphenols, including anthocyanins and flavonoids [9,10,11]. These compounds are beneficial for people because of their nutritional value. Furthermore, the bioactive compounds are known for their antioxidant, antiatherosclerotic and bactericidal properties [12,13].

The total contents of polyphenols in barley beers enhanced with Saskatoon fruits on average were 54.25% higher than in the control (CB) sample (Table 3). Saskatoon fruit is characterised by very high contents of polyphenolic compounds (3373.94 mg 100 g^−1^ d.m. for ‘Martin’ cultivar and 2689.17 mg 100 g^−1^ d.m. for ‘Smoky’ cultivar [9]). Transition of phenolic compounds contained in fruit depends on the degree of the fruit refinement. This results in breaking of the cell walls, and consequently, increased contact with the beer solution which extracts and absorbs the chemical compounds contained in the fruit [26]. Beers with addition of the red fruit of cherry dogwood (350 mg GAE/L; [34]), as well as goji berries (415 mg GAE/L; [35]), such as those assessed in this study, were found with high contents of polyphenols. On the other hand, the study by Gasiński et al. [26] reported that beer with addition of mango fruit had slightly lower contents of polyphenolic compounds, in the range of 218.6–267.6 mg GAE/L.

Polyphenolic compounds in beer were identified based on an analysis of characteristic spectral data—mass-to-charge ratio, m/z, and maximum absorption of radiation, which were compared to the available literature [9]. The characteristics of four polyphenolic compounds which were identified are shown in Table 4. The identified derivatives include derivatives of hydroxycinnamic acids (compounds 1–3) and flavan-3-ol (compound 4). Saskatoon fruits are characterised by high contents of phenolic acids, which are mainly represented by chlorogenic and neochlorogenic acid (their mean contents respectively accounted for 59% and 18% of total content of phenolic acids in Saskatoon fruit [9]. Phenolic acids are known for their antioxidant activity, they inhibit damage to DNA structure, and contribute to the aroma of fruit [9]. The contents of neochlorogenic acid in beers enhanced with untreated fruit pulp on average was 91% higher compared with beer produced with addition of ozone-treated Saskatoon fruits. On the other hand, chlorogenic acid was found only in beers enhanced with untreated Saskatoon fruits (Table 3). Catechin—a chemical compound classified in the group of flavon-3-ols—was only found in beer enhanced with ‘Amela’ fruits; furthermore, ozone treatment reduced the content of this compound in beer by approximately 59.1% (Table 3.). Catechins beneficially affect the cardiovascular system, contributing to the effective regulation of blood flow [36]. Most commonly, the content of catechin in beer is in the range of 0.03–6.54 mg/L [37]. The study by Baigts-Allende et al. [2], investigating beer enhanced with blackcurrant, reported 1.93 mg/L content of catechin, whereas caffeic acid was not detected in the samples. Derivatives of hydroxycinnamic acid include caffeic acid, which is present in all the assessed beers enhanced with fruit. The highest contents of this compound were found in beers enhanced with ‘Smoky’ fruits (SB1 and SB0), i.e., 7.20 mg/L and 5.92 mg/L, respectively. The contents of caffeic acid in the remaining fruit beers were far lower, amounting to 2.43–2.69 mg/L in the beers enhanced with untreated Saskatoon fruits and 1.86–3.96 mg/L in the samples enhanced with ozone-treated fruit. The process intended to improve storage shelf-life resulted in decreased content of caffeic acid in fruit beers, on average, by 20.62%, and in the case of beer with addition of ozone-treated ‘Amela’ fruits (AB0), the process led to an increase by 32.07% (Table 3). Caffeic acid is responsible, for instance, for blocking certain carcinogenic substances, such as nitrosamine; it also inhibits oxidation of lipoproteins and LDL-cholesterol [38]. Most commonly, the content of caffeic acid in beer is in the range of 0.00–23.50 mg/L [37].

According to many researchers [39,40,41,42,43], ozone treatment applied to fruit positively affects the contents of polyphenols (their concentrations increase in ozone-treated fruit). On the other hand, the present study showed lower concentrations of polyphenols in beers enhanced with ozone-treated fruit compared to beer produced with addition of untreated Saskatoon fruits. This may be related to additional chemical transformations during the fermentation of beers enhanced with ozone-treated fruit (interaction between ozone left on the fruit and products of fermentation process), which lead to reduced contents of the polyphenols in question, yet, this process requires further study.

The findings related to the antioxidant potential of beers enhanced with fruit of three Saskatoon cultivars, measured using DPPH^•^, FRAP and ABTS^+•^ methods, are shown in Table 5.

Beer contains antioxidant compounds, which are mainly represented by polyphenols, as well as vitamins, melanoidins and bitter acids [44,45]. Ditrych et al. [45] observed that higher contents of polyphenols in beers positively affect antioxidant potential of beer, determined with DPPH and FRAP assays. Assessment performed using the DPPH method showed higher antioxidant activity in barley beers enhanced with Saskatoon fruits (with the exception of beer produced with addition of fruit of the ‘Amela’ cultivar (AB1; Table 5). Furthermore, ozone treatment applied to the fruit led to increase in antioxidant potential of the investigated beers. Assessment performed using the FRAP method showed higher antiradical capacity in beers enhanced with fruit which were not treated with ozone. The antioxidant activity measured using the ABTS method was similar in the investigated samples of fruit beer; slightly higher antioxidant potential was only observed in beer produced with addition of ozone-treated fruit of the ‘Smoky’ cultivar—SB0 (Table 5). Saskatoon fruit, which in the present study was applied as a beer enhancer, is characterised by very high antioxidant activity, amounting to 17 mmol 100 g^−1^ d.m. in the ‘Martin’ cultivar, 20 mmol 100 g^−1^ d.m. in the ‘Smoky’ cultivar (according to FRAP assay), as well as 22 mmol 100 g^−1^ d.m. in the ‘Martin’ cultivar and 27 mmol 100 g^−1^ d.m. in the ‘Smoky’ cultivar (according to ABTS^+•^ assay; [9]).

### 2.4. Sensory Analysis of Fruit Beers

Sensory qualities of barley beer produced with addition of fruit determine the specific beer style, and contribute to the attractiveness of the beverage and the acceptance of the given type of beer among consumers. Sensory assessment of the beers produced with addition of Saskatoon fruits (ozone-treated and untreated) was carried out by a panel of ten experts, and the results are shown in Figure 4, Figure 5 and Figure 6.

The sensory profile of the investigated fruit beers was varied, which was confirmed by physicochemical analyses. Sensory assessment of the barley beers enhanced with Saskatoon fruits, treated with ozone or untreated, showed that the beer enhanced with ‘Smoky’ fruits (SB0 and SB1) had the most balanced taste profile. Significant quality attributes of that beer included intensity, perceived bitter flavour, as well as fruity taste and aroma, resulting from the addition of Saskatoon fruit (Figure 5). The remaining fruit beers investigated in the study greatly differed in terms of quality attributes, relative to the variety and enhancement of the barley beer with fruit treated with ozone (Figure 4, Figure 5 and Figure 6). In all the investigated beers, it was possible to taste a malty flavour, which is produced by compounds such as maltol and furaneol [46]. Chemical compounds significantly affecting the taste of beer are produced as a result of interactions between carbonyls, esters, sulphur compounds, alcohols, phenolic compounds and organic acids [46]. A less intense bitter flavour was identified in beers enhanced with ozone-treated fruit, compared with the samples produced with untreated Saskatoon fruits. This was linked to lower proportional contents (in the former samples) of polyphenols, i.e., caffeic acid and chlorogenic acid, whose concentration in the range of 1–10 mg/L leads to an acidic, bitter and pungent taste of beer [28]. Beer with a distinct fruity flavour, sweet aftertaste and pleasant aroma is more favoured by and desirable for consumers, compared to traditional types of beer [47,48].

## 3. Materials and Methods

### 3.1. Material

The research material comprised Saskatoon fruits of three cultivars: two Canadian cultivars (‘Martin’ and ‘Smoky’) and a new Polish cultivar (‘Amela’) (source: National Institute for Horticultural Research (InHort), Skierniewice, Poland). Two-kilogram samples of ripe fruit of each cultivar were collected manually, in early July 2021, from 6-year-old shrubs cultivated in a field experiment at the Experimental Orchard in Dąbrowice (51.9163° N/20.1009° E), belonging to the National Institute for Horticultural Research (InHort) in Skierniewice, central Poland. After they were transferred to the laboratory of the Department of Agricultural and Food Engineering at the University of Rzeszów, the fruit were divided into two samples, 1 kg each (one part was left unprocessed, and the other part was treated with ozone). The treated and the untreated samples of fruit were promptly chilled and kept in a freezer (at −18 °C) until they were used in the production of fruit beers.

The input material used in the brewing process (100%) comprised commercially available barley malt acquired from Viking Malt company in Strzegom (Poland). The barley malt had the following characteristics: extract potential—80.0% d.m., total protein content—11.4% d.m., content of soluble protein—3.75% d.m., diastatic power—324 WK, and degree of final attenuation—82.1%. The malt was refined using a Cemotec disc mill manufactured by FOSS (Sweden).

### 3.2. Ozonation Process

Saskatoon fruits in the amount of 1 kg were evenly arranged on a sieve in the middle part of a plastic container with dimensions L × W × H—0.6 × 0.4 × 0.4 m and treated to ozonation for 22 min, at ozone concentration of 10 ppm, flow rate of 4 m^3^ h^−1^, and temperature of 20 °C). The ozone was produced using TS 30 ozone generator (Ozone Solution, Hull, MA, USA) with 106 M UV Ozone Solution detector (Ozone Solution, Hull, MA, USA).

### 3.3. Production of Beer

Production of seven samples of barley beer, based on the infusion method, was carried out in the laboratory of the Department of Agricultural and Food Engineering at the University of Rzeszów. A 4.63 kg sample of refined barley malt was placed in the brew kettle ROYAL RCBM-40N (Expondo; Poland; applied at 80% process efficiency) with 13.9 L of water (3 L of water per each kilogram of malt). The mashing process was carried out for 60 min, at 67 °C (the material was vigorously stirred during the first 30 min), and then the process temperature was increased to 72 °C and maintained for 15 min. At the next stage, the temperature was increased to 78 °C for 10 min. After an iodine–starch test showed a negative result, the mashing process was completed, and the mash was subjected to filtration and sparging with water at 78 °C.

Following the latter process, the sweet wort was placed in the brew kettle ROYAL RCBM-40N and heated up to a temperature of 100 °C, at a rate of 2 °C/1 min. The wort was boiled for 60 min. During this process, hops were added, as follows: the first dose after the wort reached the boiling temperature (at 0 min)—10 g of Amarillo hops (USA; α-acid contents of 9.5%); the second dose after 45 min of boiling—5 g of Amarillo hops; and the third dose after 60 min of boiling—10 g of Amarillo hops (a source of aroma).

Subsequently, the hot wort was chilled using a spiral immersion cooler and tap water as a cooling agent. After 35 min, the temperature of the wort decreased to 20 °C. All of the seven wort samples had extract content of 12.6 °P. The cooled wort was poured in 30 l fermentation vessels. Subsequently, the yeast *Saccharomyces cerevisae* Safale US-05 (6 × 10^9^/g), earlier subjected to a dehydration process in line with the manufacturer’s instructions (0.58 g d.m./L of wort), was added to the vessels, as well. The fermentation process was carried out at 21 °C. After the fermentation process had continued for 7 days, Saskatoon fruit pulp was added to the beer, at an amount of 1 kg (21.6% relative to the weight of malt), and then the fermentation process continued for the next 14 days. After 21 days, aqueous solution of sucrose (0.3%) was added, and the beer was poured into bottles for refermentation to achieve an adequate level of carbonation. The beer was then stored at 20 °C. Sensory assessment and physicochemical tests were performed one month after the bottling.

The barley beer produced with addition of ‘Smoky’ not subjected to ozone treatment is marked as SB1, and the sample produced with addition of ozone-treated fruit of this cultivar is marked as SB0. The barley beer produced with addition of the fruit of the ‘Martin’ cultivar, not treated with ozone, is marked as MB1, and the sample produced with addition of ozone-treated fruit of the same cultivar is marked as MB0. The barley beer produced with addition of ‘Amela’ not subjected to ozone treatment is marked as AB1, and the sample produced with addition of ozone-treated fruit of the same cultivar is marked as AB0. The barley beer sample produced with no addition of Saskatoon fruit is marked as CB (Control Beer).

### 3.4. Analysis of Beer Quality Indicators

Alcohol content, apparent extract, real extract, original extract in beer, degree of final apparent attenuation and real attenuation, as well as energy value of beer, were determined using a near-infrared (NIR) spectroscopy method according to PB-ZO/PPS 16, 6th edition, from 31 May 2021 [49]. The colour of beer was assessed using the calorimetric technique defined in PN-A-79093-5:2000 [50]. The total acidity of beer was defined using the potentiometric titration method, in line with PB-ZO/PPS 18, 3rd edition, from 6 August 2020 [51]. The value of pH in the beer was assessed using the potentiometric method in accordance with PN-A-79093-4:2000 [52]. The content of carbon dioxide, CO_2_, in beer was examined using the pressure method in line with PN-A-79093-6:2000 [53]. The content of bitter substances in beer was determined using the spectrophotometric method in line with PB-ZO/PPS 10, 6th edition, from 6 August 2020 [54]. The contents of fermentation side-products in beer were determined using a gas chromatograph (Hewlett Packard 6890) with a function of flame ionization detection (HS-GC/FID), a technique complying with 9.39 Analytica EBC [55]. The analyses were performed in three replications.

### 3.5. Contents of Bioactive Compounds in Fruit Beers

#### 3.5.1. Total Contents of Polyphenols

The total contents of polyphenols in the beers were determined using the Folin–Ciocalteu (F–C) method, as described by Prior et al. [56]. For this purpose, 0.1 mL of beer was sampled, and 0.2 mL of the F–C reagent was added. After 3 min, 1 mL of 20% Na_2_CO_3_ was added along with 2 mL of distilled water. The measurement was performed after 1 h, using a spectrophotometric method at a wavelength of λ = 765 nm, with UV-Vis V-5000 spectrophotometer (Shanghai Metash Instruments Co., Ltd., Shanghai, China). The results were expressed as an equivalent of gallic acid (mmol GAE/L). The analyses were performed in three replications.

#### 3.5.2. Determination of Polyphenols Profile by UPLC-TQD-MS

Determination of polyphenolic compounds was carried out using the UPLC equipped with a binary pump, column and sample manager, photodiode array detector (PDA), tandem quadrupole mass spectrometer (TQD) with electrospray ionization (ESI) source working in negative mode (Waters, Milford, MA, USA) according to the method of Żurek et al. [57]. Separation was performed using the UPLC BEH C18 column (1.7 µm, 100 mm × 2.1 mm, Waters) at 50 °C, at a flow rate of 0.35 mL/min. The injection volume of the samples was 5 µL. The mobile phase consisted of water (solvent A) and 40% acetonitrile in water, *v*/*v* (solvent B). The following TQD parameters were used: capillary voltage of 3500 V; con voltage of 30 V; con gas flow, 100 L/h; source temperature, 120 °C; desolvation temperature, 350 °C; and desolvation gas flow rate of 800 L/h. Polyphenolic identification and quantitative analyses were performed on the basis of the mass-to-charge ratio, retention time, specific PDA spectra, fragment ions and comparison of data obtained with commercial standards and literature findings. The analyses were performed in three replications.

#### 3.5.3. Antioxidant Activity

##### DPPH Test

The antiradical activity of fruit beers was determined using the DPPH radical, in accordance with the method described by He et al. [58]. A 0.05 mmol/L solution of DPPH (2,2-diphenyl-1-picrylhydrazyl) in ethanol was prepared for this purpose. A 7.8 mL sample of the solution was placed in a test tube, with 0.2 mL of diluted (2×) beer, and incubated in darkness for 60 min at 37 °C; subsequently, the absorbance at the wavelength λ = 517 nm was examined using a UV-Vis V-5000 spectrophotometer (Shanghai Metash Instruments Co., Ltd., Shanghai, China). The control contained distilled water rather than beer. The results were expressed as Trolox equivalent (mmol TE/L). The analyses were performed in three replications.

##### FRAP Test

The reducing power of fruit beers was determined using the FRAP reagent, in accordance with the method described by Benzie and Strain [59] and He et al. [58]. The materials prepared for this purpose included a 10 mmol/L TPTZ (2,4,6-tripyridyl-s-triazine) solution, a 20 mmol/L FeCl_3_.6H_2_O solution, an acetate buffer with pH = 3.6, as well as a 40 mmol/L HCl solution. Subsequently, the FRAP reagent was prepared by mixing 25 mL of the acetate buffer with 2.5 mL of the TPTZ dissolved in HCl and 2.5 mL of FeCl_3_.6H_2_O. A 6 mL sample of FRAP solution was placed in a test tube with 0.2 mL of the beer and incubated at a temperature of 37 °C for 10 min; subsequently, the absorbance at the wavelength of λ = 593 nm was examined using a UV-Vis V-5000 spectrophotometer (Shanghai Metash Instruments Co., Ltd., Shanghai, China). The control contained distilled water instead of beer. The results of the FRAP test were expressed as mmol Fe^2+^/L. The analyses were performed in three replications.

##### ABTS Test

Antiradical activity of fruit beers was examined using ABTS radical cation in line with the method proposed by Re et al. [60]. A 7 mmol/L ABTS (2,2′-azinobis(3-ethylbenzothiazoline-6-sulfonic acid) solution and a 2.45 mmol/L potassium persulphate solution were prepared for this purpose. The solutions were combined at 1:0.5 ratio, and stored for 12–16 h in darkness to enable the development of ABTS cation. The ABTS^+^ solution was diluted with distilled water to achieve absorbance of 0.700 ± 0.002 (at wavelength λ = 734 nm). A 3 mL portion of the diluted ABTS^+^ solution was placed in the test tube with 0.3 mL of the beer, and after 6 min, the absorbance value at the wavelength λ = 734 nm was determined using a UV-Vis V-5000 spectrophotometer (Shanghai Metash Instruments Co., Ltd., Shanghai, China). The results were corrected to account for dilution and expressed as Trolox equivalent (mmol TE/L). The analyses were performed in three replications.

### 3.6. Sensory Assessment in Beers

The sensory assessment was performed by a team of 10 experts (6 men and 4 women aged 30–40) in the sensory analysis laboratory according to the EBC Method 13.13 [61]. The encoded beer samples were cooled to 10 °C and presented to the evaluators in a random order in transparent 100 mL (50 mL beer) glasses. Between each assessment, water was administered to rinse the mouth. The sensory profile applied to the evaluated beers determines the occurrence or absence of given qualitative characteristics of the taste and aroma of fruit beer, i.e., malty, fruity, sweet, grainy, strong, full, fresh, phenolic, bitter and sour, whose definitions and formulas of flavour notes was developed on the basis of the EBC Method 13.12 [62]. They then compared the sensory profile of beer produced with addition of Saskatoon fruits of each variety with the profile identified for the control beer. The results related to the sensory profile of the beers were examined using one-way analysis of cultivation (ANOVA). The significance of the differences between the means was assessed with Tukey’s test, with the adopted significance level of 5%.

### 3.7. Statistical Analysis

The results of fruit beer assessments are shown as mean values and standard deviations. The statistical analyses of the results were computed using Statistica 13.3 (TIBCO Software Inc., Tulsa, OK, USA). The results related to the physicochemical characteristics, polyphenol contents and antioxidant activity of fruit beers were examined using the two-factor completely randomized ANOVA, with significance level α = 0.05. The mean values were compared using Tukey’s HSD test.

## 4. Conclusions

The study, designed to assess the feasibility of Saskatoon fruits as an enhancer, to be used in production of fruit beers, showed that the most balanced sensory profile (intensity, perceived bitter flavour, as well as fruity taste and aroma) is found in barley beers enhanced with ‘Smoky’ fruits, regardless of whether they had been treated with ozone. Additionally, these beer samples were found to have better colour, higher alcohol content and a greater degree of final fermentation. Ozone treatment positively affected the antioxidant activity of fruit beers, but it also led to lower contents of organic acids, i.e., caffeic, chlorogenic and neochlorogenic acids, compared to beers produced with the addition of untreated Saskatoon fruits. Addition of Saskatoon fruits, mainly of the ‘Smoky’ cultivar, irrespective of the process intended to extend storage shelf-life, can be applied to enhance the quality of fruit beers. In addition, the process of brewing beers using ozonated fruits can be easily transferred to an industrial scale, which provides opportunities for the development of this type of beer.

## Figures and Tables

**Figure 1 molecules-27-01976-f001:**
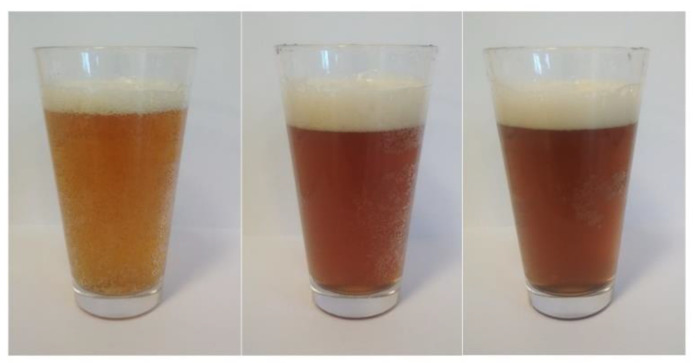
The appearance of the obtained barley beers with addition of the Saskatoon fruits; from left: CB—control beer, AB0—cv. ‘Amela’ with ozone-treated fruit and AB1—untreated fruit.

**Figure 2 molecules-27-01976-f002:**
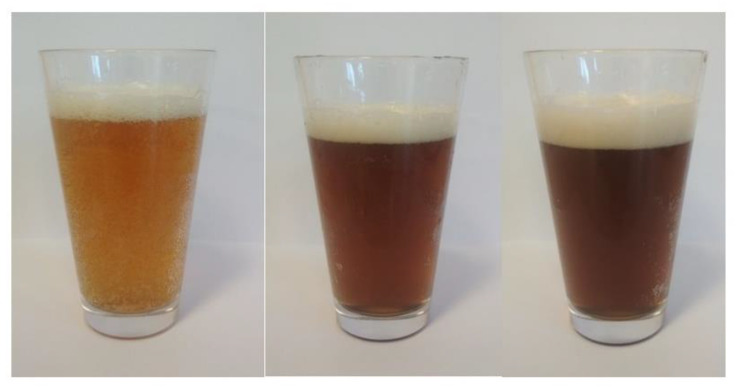
The appearance of the obtained barley beers with addition of the Saskatoon fruits. From left: CB—control, SB0—cv. ‘Smoky’ with ozone-treated fruit and SB1—untreated fruit.

**Figure 3 molecules-27-01976-f003:**
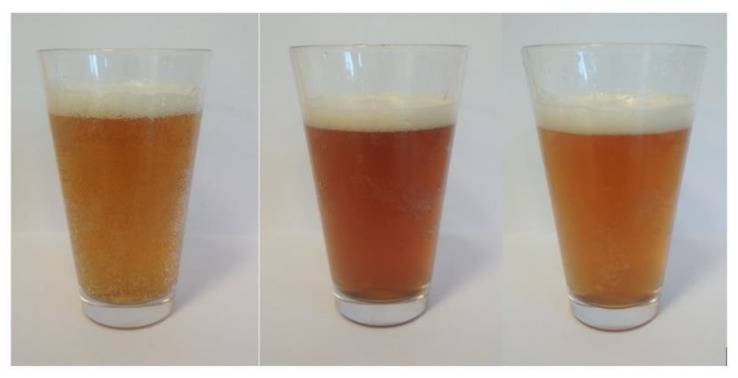
The appearance of the obtained barley beers with addition of the Saskatoon fruits. From left: CB—control, MB0—cv. ‘Martin’ with ozone-treated fruit and MB1—untreated fruit.

**Figure 4 molecules-27-01976-f004:**
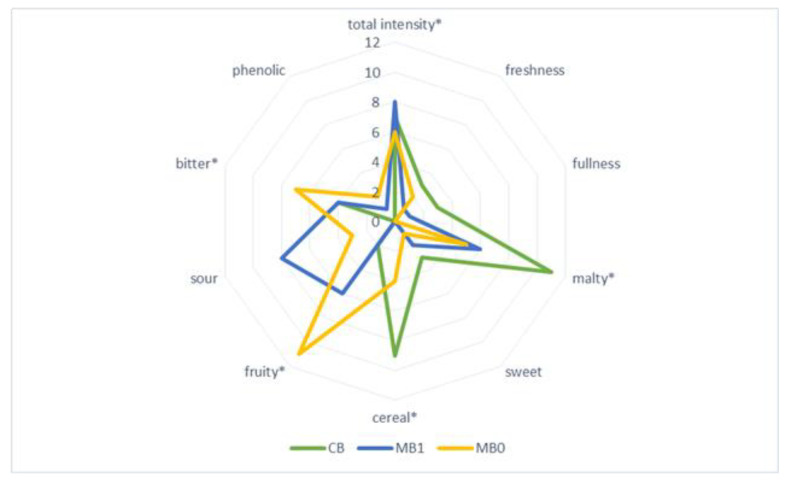
Sensory profile of barley beers—control (CB) and sample with addition of ‘Martin’ cultivar fruit untreated (MB1) and treated with ozone (MB0); * marks the attributes which were statistically different at *p* ≤ 0.05.

**Figure 5 molecules-27-01976-f005:**
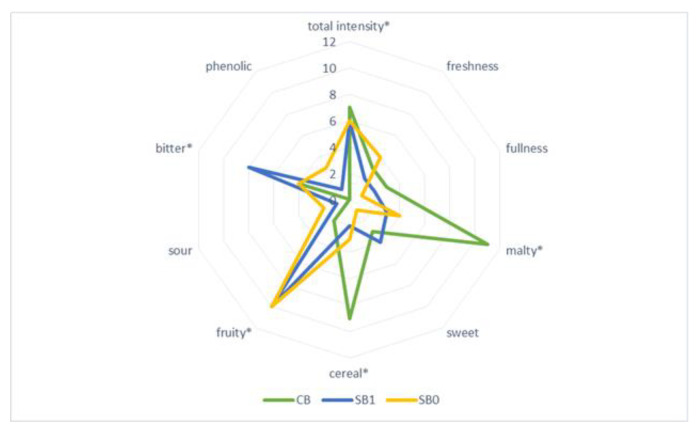
Sensory profile of barley beers—control (CB) and sample with addition of ‘Smoky’ cultivar fruit untreated (SB1) and treated with ozone (SB0); * marks the attributes which were statistically different at *p* ≤ 0.05.

**Figure 6 molecules-27-01976-f006:**
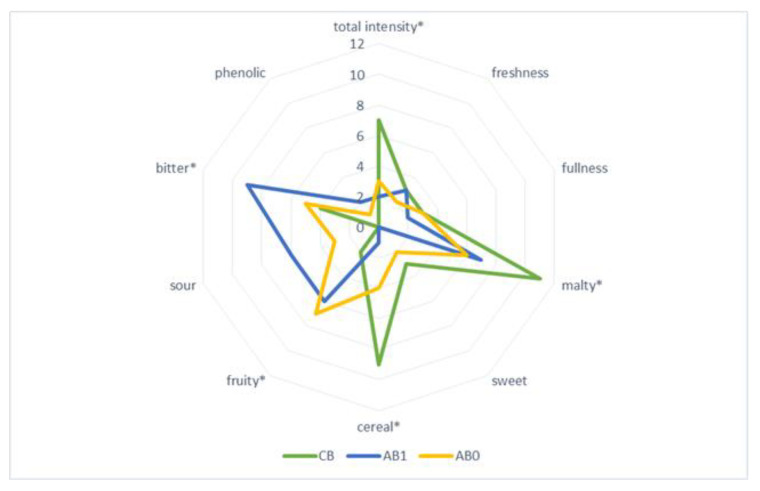
Sensory profile of barley beers—control (CB) and sample with addition of ‘Amela’ cultivar fruit untreated (AB1) and treated with ozone (AB0); * marks the attributes which were statistically different at *p* ≤ 0.05.

**Table 1 molecules-27-01976-t001:** Physicochemical analysis of the beers produced with an addition of the Saskatoon fruit.

Type of Beer	CB	AB0	AB1	SB0	SB1	MB0	MB1
Apparent extract (%; *m*/*m*)	2.74 ^cd^ ± 0.09	3.29 ^a^ ± 0.10	2.88 ^bc^ ± 0.09	2.77 ^c^ ± 0.09	2.59 ^d^ ± 0.08	3.24 ^a^ ± 0.10	3.01 ^b^ ± 0.10
Real extract (%; *m*/*m*)	4.65 ^c^ ± 0.06	4.95 ^a^ ± 0.06	4.62 ^cd^ ± 0.06	4.80 ^b^ ± 0.06	4.57 ^d^ ± 0.06	4.96 ^a^ ± 0.06	4.77 ^b^ ± 0.06
Original extract (%; *m*/*m*)	12.67 ^b^ ± 0.13	11.87 ^d^ ± 0.12	11.90 ^d^ ± 0.12	13.32 ^a^ ± 0.14	12.88 ^b^ ± 0.13	12.16 ^c^ ± 0.13	12.14 ^c^ ± 0.13
Degree of final apparent attenuation (%)	78.3 ^a^ ± 1.8	72.3 ^c^ ± 1.6	75.8 ^b^ ± 1.7	79.2 ^a^ ± 1.8	78.9 ^a^ ± 1.8	73.4 ^c^ ± 1.7	75.2 ^b^ ± 1.7
Degree of final real attenuation (%)	64.8 ^a^ ± 1.5	59.9 ^d^ ± 1.4	62.7 ^b^ ± 1.4	65.6 ^a^ ± 1.5	66.1 ^a^ ± 1.5	60.8 ^cd^ ± 1.4	62.2 ^bc^ ± 1.4
Content of alcohol (%; *m*/*m*)	4.15 ^b^ ± 0.15	3.58 ^d^ ± 0.13	3.75 ^cd^ ± 0.13	4.43 ^a^ ± 0.16	4.31 ^ab^ ± 0.15	3.72 ^cd^ ± 0.13	3.81 ^c^ ± 0.13
Content of alcohol (%; *v*/*v*)	5.30 ^b^ ± 0.14	4.58 ^d^ ± 0.12	4.80 ^c^ ± 0.13	5.66 ^a^ ± 0.15	5.51 ^a^ ± 0.15	4.76 ^c^ ± 0.13	4.87 ^c^ ± 0.13
Colour (EBC units)	18.2 ^e^ ± 0.8	20.4 ^c^ ± 0.9	23.9 ^a^ ± 1.0	24.1 ^a^ ± 1.0	24.0 ^a^ ± 1.0	21.6 ^b^ ± 0.9	19.1 ^d^ ± 0.8
Titratable acidity (0.1 M NaOH/100 cm^3^)	1.9 ^a^ ± 0.3	2.2 ^a^ ± 0.3	2.3 ^a^ ± 0.3	2.2 ^a^ ± 0.3	2.2 ^a^ ± 0.3	2.2 ^a^ ± 0.3	2.3 ^a^ ± 0.3
pH	4.58 ^ab^ ± 0.11	4.50 ^abc^ ± 0.11	4.42 ^c^ ± 0.11	4.60 ^ab^ ± 0.12	4.61 ^a^ ± 0.12	4.58 ^ab^ ± 0.11	4.46 ^bc^ ± 0.11
Content of carbon dioxide (%)	0.43 ^a^ ± 0.30	0.43 ^a^ ± 0.31	0.44 ^a^ ± 0.30	0.46 ^a^ ± 0.30	0.47 ^a^ ± 0.30	0.46 ^a^ ± 0.30	0.43 ^a^ ± 0.30
Bitter substances (IBU)	19.2 ^a^ ± 0.8	15.9 ^c^ ± 0.7	14.7 ^d^ ± 0.6	17.8 ^b^ ± 0.8	17.7 ^b^ ± 0.8	16.3 ^c^ ± 0.7	15.8 ^c^ ± 0.7
Energy value (kcal/100 mL)	46 ^a^ ± 1	43 ^a^ ± 0	43 ^a^ ± 1	48 ^a^ ± 2	47 ^a^ ± 0	44 ^a^ ± 2	44 ^a^ ± 1
Energy value (kJ/100 mL)	192 ^b^ ± 1	180 ^c^ ± 0	180 ^c^ ± 1	202 ^a^ ± 2	195 ^ab^ ± 0	184 ^c^ ± 2	184 ^c^ ± 1

Data are expressed as a mean value (*n* = 3) ± SD; SD—standard deviation. Mean values within a row with different letters are significantly different (*p* < 0.05); CB—control beer, AB—‘Amela’ cultivar, SB—‘Smoky’ cultivar, MB—‘Martin’ cultivar; 1—untreated fruit, 0—ozone-treated fruit.

**Table 2 molecules-27-01976-t002:** Contents of volatile organic compounds in beers with addition of the Saskatoon fruit.

Compound	CB	AB0	AB1	SB0	SB1	MB0	MB1
Acetic aldehyde (mg/L)	8.3 ^a^ ± 0.4	3.9 ^bc^ ± 0.2	4.3 ^bc^ ± 0.3	5.0 ^b^ ± 0.3	3.8 ^c^ ± 0.2	3.5 ^c^ ± 0.2	1.4 ^d^ ± 0.1
Ethyl acetate (mg/L)	15.8 ^e^ ± 0.5	16.6 ^de^ ± 0.5	13.6 ^f^ ± 0.3	18.8 ^c^ ± 0.3	27.6 ^b^ ± 0.2	17.1 ^d^ ± 0.1	43.9 ^a^ ± 0.0
n-Propanol (mg/L)	34.9 ^e^ ± 0.3	36.3 ^d^ ± 0.3	35.2 ^e^ ± 0.2	37.3 ^c^ ± 0.2	39.9 ^a^ ± 0.1	38.4 ^b^ ± 0.3	35.3 ^e^ ± 0.4
Isobutanol (mg/L)	74.7 ^e^ ± 0.4	78.8 ^d^ ± 0.3	84.1 ^c^ ± 0.0	99.4 ^a^ ± 0.3	79.5 ^d^ ± 0.3	94.9 ^b^ ± 0.1	84.7 ^c^ ± 0.3
Isoamyl acetate (mg/L)	0.48 ^c^ ± 0.04	0.44 ^cd^ ± 0.00	0.23 ^f^ ± 0.04	0.64 ^b^ ± 0.02	1.18 ^a^ ± 0.01	0.39 ^d^ ± 0.03	0.31 ^e^ ± 0.02
Amyl alcohols (mg/L)	89.6 ^c^ ± 0.1	88.7 ^d^ ± 0.3	87.8 ^e^ ± 0.4	95.8 ^a^ ± 0.5	91.4 ^b^ ± 0.1	89.1 ^cd^ ± 0.2	83.5 ^f^ ± 0.5
Diacetyl (mg/L)	0.015 ^c^ ± 0.004	0.025 ^b^ ± 0.004	0.023 ^b^ ± 0.003	0.023 ^b^ ± 0.001	0.021 ^bc^ ± 0.002	0.014 ^c^ ± 0.003	0.033 ^a^ ± 0.003

Data are expressed as a mean value (*n* = 3) ± SD; SD—standard deviation. Mean values within a row with different letters are significantly different (*p* < 0.05); CB—control beer, AB—‘Amela’ cultivar, SB—‘Smoky’ cultivar, MB—‘Martin’ cultivar; 1—untreated fruit, 0—ozone-treated fruit.

**Table 3 molecules-27-01976-t003:** Contents of polyphenols and polyphenolic profile of fruit beers.

Compound	CB	AB0	AB1	SB0	SB1	MB0	MB1
Total polyphenols (mg GAE/L)	176 ^e^ ± 5	373 ^cd^ ± 3	366 ^d^ ± 6	401 ^a^ ± 3	395 ^a^ ± 4	391 ^ab^ ± 1	382 ^bc^ ± 3
Neochlorogenic acid (mg/L)	n.d.	0.05 ^c^ ± 0.01	0.41 ^b^ ± 0.03	0.07 ^c^ ± 0.02	0.62 ^a^ ± 0.02	0.59 ^a^ ± 0.01	0.02 ^c^ ± 0.05
Chlorogenic acid (mg/L)	n.d.	n.d	1.16 ^a^ ± 0.00	n.d.	1.05 ^b^ ± 0.03	0.82 ^c^ ± 0.02	n.d.
Caffeic acid (mg/L)	n.d.	3.96 ^c^ ± 0.05	2.69 ^d^ ± 0.03	5.92 ^b^ ± 0.01	7.20 ^a^ ± 0.02	2.43 ^e^ ± 0.03	1.86 ^f^ ± 0.02
(+)Catechin (mg/L)	n.d.	1.03 ^b^ ± 0.03	2.52 ^a^ ± 0.06	n.d.	n.d.	n.d.	n.d.

n.d.—concentration not detected; Data are expressed as a mean value (*n* = 3) ± SD; SD—standard deviation. Mean values within a row with different letters are significantly different (*p* < 0.05); CB—control beer, AB—‘Amela’ cultivar, SB—‘Smoky’ cultivar, MB—‘Martin’ cultivar; 1—untreated fruit, 0—ozone-treated fruit.

**Table 4 molecules-27-01976-t004:** Polyphenolic profile identified by UPLC-PDA-TQD-MS.

	Compound	R_t_ (min)	MS (*m*/*z*)	MS/MS (*m*/*z*)	λ_max_ (nm)
1.	Neochlorogenic acid	2.84	353	191	299sh, 327
2.	Chlorogenic acid	2.96	353	191	299sh, 327
3.	Caffeic acid	3.19	179	-	299sh, 324
4.	(+)Catechin	3.40	289	-	273

R_t_—retention time.

**Table 5 molecules-27-01976-t005:** Antioxidant potential of fruit beers with Saskatoon fruit pulp added.

Type of Beer	CB	AB0	AB1	SB0	SB1	MB0	MB1
DPPH^•^ (mmol TE/L)	1.42 ^cd^ ± 0.08	1.52 ^bcd^ ± 0.08	1.38 ^d^ ± 0.08	1.80 ^a^ ± 0.08	1.68 ^ab^ ± 0.06	1.56 ^bc^ ± 0.09	1.58 ^bc^ ± 0.10
FRAP (mmol Fe^2+^/L)	1.15 ^e^ ± 0.03	1.63 ^d^ ± 0.06	1.83 ^c^ ± 0.04	2.28 ^b^ ± 0.08	2.46 ^a^ ± 0.07	1.62 ^d^ ± 0.05	1.81 ^c^ ± 0.07
ABTS^+•^ (mmol TE/L)	1.60 ^a^ ± 0.13	1.66 ^a^ ± 0.06	1.65 ^a^ ± 0.06	1.80 ^a^ ± 0.08	1.63 ^a^ ± 0.10	1.64 ^a^ ± 0.06	1.67 ^a^ ± 0.07

Data are expressed as a mean value (*n* = 3) ± SD; SD—standard deviation. Mean values within a row with different letters are significantly different (*p* < 0.05); CB—control beer, AB—‘Amela’ cultivar, SB—‘Smoky’ cultivar, MB—‘Martin’ cultivar; 1—untreated fruit, 0—ozone-treated fruit.

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
