# Peer review of "Effect of Ozone-Treated or Untreated Saskatoon Fruits (Amelanchier alnifolia Nutt.) Applied as an Additive on the Quality and Antioxidant Activity of Fruit Beers"

_molecules, 2022, doi:10.3390/molecules27061976_

Round 1
Reviewer 1 Report
The objective of the work is sound and novel. Although results with ozone treated fruits are not significantly attractive, the overall work could be interesting to most of the readers. The presentation and discussion of results are quite sound.
Author Response
The Authors are grateful for the contribution of the Reviewer.
According to the Reviewer's comments, the manuscript has been revised by a native speaker.
Reviewer 2 Report
Effect of ozone_treated or untreated Saskatoon fruits (Amelanchier alnifolia Nutt) applied as an additive, on the quality and antioxidant activity of fruit beers
This paper describes the benefits of the contribution of Saskatoon fruit on the quality of a beer when the fruit is treated with an ozone treatment and the antioxidant contribution to the final product when Saskatoon fruit is added.
Mayor concern
In general, the work is very well described and is interesting from the point of view of the organoleptic improvement that the addition of a fruit such as Saskatoon to the brewing process entails, as well as the antioxidant properties that this fruit would provide to the beverage. In addition, the benefit that the ozonation process generates to the fruit. However, in my opinion, there are some aspects that could improve the present work, which I mention below.
In the abstract of this work, gaseous ozone treatment is mentioned, but the treatment conditions should be further elaborated. Please, could you include something more?
The introduction is quite complete and well referenced. However, few explanations and references are devoted to the ozonation process. Could you expand more on the ozonation process and the improvement it brings to the brewing industry or even to other industries?
The present study does not say anything about the influence of ozone treatment on microbial activity, which is responsible for extending the shelf life of the final product, as mentioned in the introduction and even in the final conclusions of the work. Do you think it would be interesting to carry out a microbiological analysis to extend the study?
In the materials and methods section, an item referring to ozone treatment should be included separately from the raw material used in the study. See line 320.
The work should take into account and clarify whether the brewing process using ozone-treated fruit would be easy or possible to take to industrial scale and whether the research team has considered the possibility of improving the system to be protected under patent.
Minor concern
Line 323: Change to 20ºC
Line 326: should write “freezer” instead of “refrigerator”.
Line 336: What is meant by the infusion method?
Line 347: 2 ºC (C is missing).
Line 423: 0.2 mL instead of m3
Line 451: 0.700±0.02… there is one more decimal place.
Author Response
The Authors are grateful for the contribution of the Reviewer.
In the abstract of this work, gaseous ozone treatment is mentioned, but the treatment conditions should be further elaborated. Please, could you include something more?
Answer:
Saskatoon fruits in the amount of 1 kg were evenly arranged on a sieve in the middle part of the plastic container with dimensions of L x W x H (0.6 x 0.4 x 0.4 m) and treated to ozonation for 22 minutes, at ozone concentration of 10 ppm, flow rate of 4 m3.h-1, and temperature of 20°C). The ozone was produced using TS 30 ozone generator (Ozone Solution, Hull, MA, USA) with 106 M UV Ozone Solution detector (Ozone Solution, Hull, MA, USA). (lines 357 – 361).
The introduction is quite complete and well referenced. However, few explanations and references are devoted to the ozonation process. Could you expand more on the ozonation process and the improvement it brings to the brewing industry or even to other industries?
Answer:
Studies have shown that the use of ozone in gaseous form is more effective in relation to the aqueous form of ozone (faster process of ozone breakdown in water). By applying ozone, it is possible to limit the development of various fungal diseases, such as gray mould (Botrytis cinerea), which has an impact on extending the shelf life of fruits, especially those used in food processing (ozone can be used at any stage - immediately after harvest, during transport, sorting or packaging). Ozonation leads to improve processing properties ( including reduces ethylene secretion, increases polyphenol content and antioxidant activity in fruits, and causes modifications in the activity of enzymes found in vegetables) and microbiological safety of food [20-24]. Fruits have microorganisms on their surface, especially yeasts and molds, which during fermentation pass into young beer, causing undesirable processes that significantly affect both the appearance of beer (color) as well as taste sensations (e.g. the formation of diacetyl or phenols). The applied process of ozonation of fruits before their addition to beer can inhibit the growth of microorganisms affecting the microbiological stability of the finished product. (lines 75-88).
The present study does not say anything about the influence of ozone treatment on microbial activity, which is responsible for extending the shelf life of the final product, as mentioned in the introduction and even in the final conclusions of the work. Do you think it would be interesting to carry out a microbiological analysis to extend the study?
Answer:
The study of the microbiological activity of beers is very interesting, but due to the extensiveness of the research, we have not performed tests of the microbiological activity of beers. At the same time, beers were made on a laboratory scale and were not preserved in any way, e.g. by pasteurization or filtration, therefore microbiological activity could change quickly, especially considering the undisturbed activity of yeast and other microorganisms (additionally supported by the addition of sugar to refermentation in bottles). Of course, we plan further research on the microbiological activity of wort, added fruit and the finished beer product.
In the materials and methods section, an item referring to ozone treatment should be included separately from the raw material used in the study. See line 320.
Answer:
In the material and methods section, a separate ozonation process point (3.2) has been added (line 357).
The work should take into account and clarify whether the brewing process using ozone-treated fruit would be easy or possible to take to industrial scale and whether the research team has considered the possibility of improving the system to be protected under patent.
Answer:
The industrial process of producing fruit beers using ozonated fruit would require the purchase of an ozone generator, which is a relatively high cost, while when it comes to introducing an additional device (ozone generator) on the technological line, this should not be a problem. Fruit ozonation processes and their impact on the quality of the raw material are widely known, while the technology of production of beers using ozonated fruits is possible to develop patent protection. (lines 519-521).
Minor concern
Line 323: Change to 20ºC - it is corrected
Line 326: should write “freezer” instead of “refrigerator”. - it is corrected
Line 336: What is meant by the infusion method?
Answer:
In brewing, two mashing methods are used – the infusion method and the decoction method. The difference lies in the way the mash is heated. In the infusion method, the mash is heated at a certain rate (usually 1-2°C/min) to the total saccharification temperature, i.e. 76 - 78°C, using periodic breaks for prolonged action of enzymes.
Line 347: 2 ºC (C is missing). - it is corrected
Line 423: 0.2 mL instead of m3 - it is corrected
Line 451: 0.700±0.02… there is one more decimal place. - it is corrected
Reviewer 3 Report
The article "Effect of ozone-treated or untreated Saskatoon fruits (Amelanchier alnifolia Nutt.) applied as an additive, on the quality
and antioxidant activity of fruit beers" presents an interesting study that requires some improvements to be published in Molecules:
- Please include more information in the introduction section about ozone, its form of application and its effects on the physicochemical properties of fruits potentially considered for beer production.
- Please include in materials and methods more details about the sensory evaluation, in particular the test used and the bibliographic references (international standards) considered.
- Please include some images of the samples so that readers can observe the products under evaluation.
- Please discuss in relation to the color of the samples and if ozone affects this important physical property.
Author Response
The Authors are grateful for the contribution of the Reviewer.
According to the Reviewer's comments, the manuscript has been revised by a native speaker.
- Please include more information in the introduction section about ozone, its form of application and its effects on the physicochemical properties of fruits potentially considered for beer production.
Answer:
Studies have shown that the use of ozone in gaseous form is more effective in relation to the aqueous form of ozone (faster process of ozone breakdown in water). By applying ozone, it is possible to limit the development of various fungal diseases, such as gray mould (Botrytis cinerea), which has an impact on extending the shelf life of fruits, especially those used in food processing (ozone can be used at any stage - immediately after harvest, during transport, sorting or packaging). Ozonation leads to improve processing properties ( including reduces ethylene secretion, increases polyphenol content and antioxidant activity in fruits, and causes modifications in the activity of enzymes found in vegetables) and microbiological safety of food [20-24]. Fruits have microorganisms on their surface, especially yeasts and molds, which during fermentation pass into young beer, causing undesirable processes that significantly affect both the appearance of beer (color) as well as taste sensations (e.g. the formation of diacetyl or phenols). The applied process of ozonation of fruits before their addition to beer can inhibit the growth of microorganisms affecting the microbiological stability of the finished product. (lines 75-88).
- Please include in materials and methods more details about the sensory evaluation, in particular the test used and the bibliographic references (international standards) considered.
Answer:
The sensory assessment was performed by a team of 10 experts (6 men and 4 women aged 30 - 40) in the sensory analysis laboratory according to ECB Method 13.13 [61]. The encoded beer samples were cooled to 10°C and given to the evaluators in random order in transparent 100 mL (50 mL beer) glasses. Between each assessment, water was administered to rinse the mouth. The sensory profile applied to the evaluated beers determine the occurrence or absence of given qualitative characteristics the taste and aroma of fruit beer, i.e., malty, fruity, sweet, grainy, strong, full, fresh, phenolic, bitter and sour whose definitions and formulas of flavour notes was developed on the basis of ECB Method 13.12 [62]They then compared the sensory profile of beer produced with the addition of Saskatoon fruits of each cultivar with the profile identified for the control beer. (lines 488-497).
- Please include some images of the samples so that readers can observe the products under evaluation.
Answer:
There are pictures of beers on page 4 of the manuscript.
- Please discuss in relation to the color of the samples and if ozone affects this important physical property.
Answer:
Both the dose used and the exposure time of ozone affect the colour of the fruits undergoing the process. Ozonated fruits retain their natural colour longer in relation to non-ozonated fruits, which darken faster [20-24]. Already at the stage of preparing of the Saskatoon fruit pulp, a difference in colour between ozonated and non-ozonated fruits within the cultivar was noticed, which had an impact on the final color of the resulting fruit beers. (lines 148-153).
Round 2
Reviewer 2 Report
The present article proposed for publication in the journal Molecules has been satisfactorily revised by the authors. From my point of view all the issues I raised in the first review have been answered and expanded in the manuscript. Therefore, I consider the present work suitable for publication in your journal. Thank you very much.
Best regards.
Reviewer 3 Report
Accept in present form